

# The influence of ecological infrastructures adjacent to crops on their carabid assemblages in intensive agroecosystems

Emilie Pecheur[1,2], Julien Piqueray[3], Arnaud Monty[1,2], Marc Dufrêne[1,2] and Grégory Mahy[1,2]

[1] Gembloux Agro-Bio Tech, Biodiversity and Landscape, University of Liège, Gembloux, Belgium
[2] Gembloux Agro-Bio Tech, TERRA-AgricultureIsLife, University of Liège, Gembloux, Belgium
[3] Natagriwal asbl, Gembloux, Belgium

## ABSTRACT

**Background**. Conserving biodiversity and enhancing ecosystem services of interest in intensive agroecosystems is a major challenge. Perennial ecological infrastructures (EIs), such as hedges and grassy strips, and annual EI under Agri-Environment Schemes appear to be good candidates to promote both. Our study focused on carabids, an indicator group responding both at the species and functional trait level to disturbances and supporting pest control and weed seed consumption services.

**Methods**. We compared carabid assemblages at the species and functional traits levels, sampled via pitfall trapping, in three types of EIs (hedges, grassy strips and annual flower strips) and crops. We also tested via GLMs the effect of (1) the type of EI at the crops' border and (2) the distance from the crops' border (two meters or 30 meters) on carabid assemblages of crops. Tested variables comprised: activity-density, species richness, functional dispersion metrics (FDis) and proportions of carabids by functional categories (Diet: generalist predators/specialist predators/seed-eaters; Size: small/medium/large/very large; Breeding period: spring/autumn).

**Results and Discussion**. Carabid assemblages on the Principal Coordinate Analysis split in two groups: crops and EIs. Assemblages from all sampled EIs were dominated by mobile generalist predator species from open-land, reproducing in spring. Assemblages of hedges were poor in activity-density and species richness, contrarily to grassy and annual flower strips. Differences in carabid assemblages in crops were mainly driven by the presence of hedges. The presence of hedges diminished the Community Weighted Mean size of carabids in crops, due to an increased proportion of small (<5 mm) individuals, while distance from crops' border favoured large (between 10–15 mm) carabids. Moreover, even if they were attracted by EIs, granivorous carabid species were rare in crops. Our results underlie the importance of local heterogeneity when adapting crops' borders to enhance carabid diversity and question the relevance of hedge implantation in intensive agrolandscapes, disconnected from any coherent ecological network. Moreover, this study emphasizes the difficulty to modify functional assemblages of crops for purposes of ecosystem services development, especially for weed seed consumption, as well as the role of distance from the crops' border in the shaping of crop carabid assemblages.

Corresponding author
Emilie Pecheur,
epecheur@doct.uliege.be

## INTRODUCTION

It is well acknowledged that intensive agriculture, the dominating production model in Europe, put farmland biodiversity in jeopardy (*Donald, Green & Heath, 2001*; *Stoate et al., 2001*). This raises concerns since biodiversity has been linked to the efficient and durable delivery of ecosystem services in agroecosystems. One of the most studied taxa in agroecosystems, both for its contribution to ecosystem services and as an indicator group (*Holland & Luff, 2000*), are carabids. Carabids respond to habitat changes and disturbances (*Brooks et al., 2012*), causing shifts in community assemblages at the species and functional traits levels (*Hanson et al., 2016*; *Kotze & O'Hara, 2003*). From an agronomic point of view, they are potential providers of two ecosystem services of interest in agroecosystems: pest control (*Kromp, 1999*) and weed seeds removal (*Bohan et al., 2011*).

Preserving carabid diversity in agrolandscapes goes along with the implementation of a green network of ecological infrastructures (EIs). Indeed, carabid communities appear to benefit from spatial landscape heterogeneity (*Neumann et al., 2016*). Increased habitat type diversity, as well as the presence of linear features varying in length and composition, have positive effects on carabid community compositions (*Duflot et al., 2016*; *Duflot et al., 2017*; *Fahrig et al., 2015*; *Woodcock et al., 2005*). Conservation of source habitats must be planned, such as forested patches and permanent grasslands (*Purtauf et al., 2005*), and completed by a network of less disturbed perennial linear elements adjacent to fields, such as hedges and grassy strips (*Fournier & Loreau, 2001*). Potentially acting as connection elements, allowing species to spread into the fields (*Hof & Bright, 2010*; *Labruyere et al., 2016*; *Labruyere, Petit & Ricci, 2017*), perennial linear elements are also of great importance to preserve locally biodiversity in the fields, notably by providing refuge areas or overwintering sites (*Fournier & Loreau, 2001*; *MacLeod et al., 2004*).

In the European Union, perennial elements as well as a variety of annual flower strips can be established under the framework of Agri-Environment Schemes (AES). Different seed mixes, aiming at various environmental purposes, are sown each year in a ploughed strip along crop borders. Less disturbed than crops, since chemical inputs are prohibited, these strips could attract a variety of border-specific carabids species (*Saska et al., 2007*). When sown along a perennial EI, annual flower strips could also enhance carabid diversity, thanks to the additive effect of both habitats (*Boetzl, Schneider & Krauss, 2016*).

The successful establishment of communities of arthropods in EIs is largely constrained by local and landscape scales (*Gabriel et al., 2010*). The potential of AES regarding biodiversity support in impoverished agrolandscapes may thus be limited (*Kleijn et al., 2006*), a fact underlying the theory that AES will perform best in moderately complex agrolandscapes (*Concepción et al., 2012*). Still, intensive agrolandscapes, either having undergone radical landscape simplification or being historically intensive (i.e., run under an open-field regime), represent a large share of the potential areas where biodiversity could benefit from the implementation/conservation of EIs. Several studies have explored carabid assemblages in agrolandscapes dominated by a bocage pattern undergoing a loss of connectivity and a consequent fragmentation (*Baudry, Bunce & Burel, 2000*; *Fournier & Loreau, 2001*; *Neumann et al., 2016*). Regarding the influence of crops on carabids (*Eyre,*

*Luff & Leifert, 2013*), historically intensive agroecosystems are likely to have different carabid beetles assemblages.

Addressing ecological functionality of carabid assemblages in intensive agroecosystems requires to consider the diversity of functional traits within the communities. The interest of integrating functional traits to the usual diversity indices is twofold. First, species richness alone is neither the best explanation nor the insurance of ecosystem functioning (*Cadotte, Carscadden & Mirotchnick, 2011*). Second, landscape characteristics shape communities by a filtering operating on their functional traits (*De Palma et al., 2015*; *Duflot et al., 2014b*). Eventually, this filtering applies a selection pressure on defined functional groups and may lead to a homogenisation of communities (*Barbaro & Van Halder, 2009*). Therefore, functional traits facilitate the inclusion of landscape history as a factor shaping existing species assemblages, easing the generalisation of results (*Gagic et al., 2015*).

Local planning, however, must not be neglected in the mitigation of biodiversity erosion in agroecosystems and the maintenance of ecosystem services. Therefore, this study focusing on historically intensive agrolandscapes aims at (1) comparing carabid assemblages and functional groups in three types of EIs: grassy strips (perennial), hedges (perennial) and annual flower strips to evaluate the potential of these EIs in biodiversity conservation; (2) testing how the type of EI can modify crop assemblages of carabids at the field border and at 30 m, for purposes of promoting pest and weed seed control.

## MATERIALS & METHODS

Three study sites were selected in southern Belgium (Figs. 1A, 1B). Sampled fields were all flat, shared the same biogeographical context and were situated at the same altitude. In ArcMap 10.2, we calculated for each crop the cover percentage of several land use categories (see Annex 1) within radii of 50 and 500 m, distances known to influence carabid assemblages (*Aviron et al., 2005*). Data for land cover was provided by the shapefile TOP10V (©IGN, 2018). Located on loamy soils, sites were all intensively managed, surrounded by at least 75% of cropped area at 50 and 500 m (Annex 1). Percentage of agricultural area of ecological potential, i.e., set of (semi-) natural elements present on arable lands, was comprised between 1.5 and 5% (*Piqueray et al., 2013*). Eventually, an ANOSIM confirms that carabid communities from the crops were similar between the three study sites ($p$-value $= 0.086$, $R = 0.063$) (*Clarke & Warwick, 2001*). This is corroborated with results from homogeneity of groups dispersion, tested with 'betadisper' from the 'vegan' package (*Oksanen et al., 2008*) (ANOVA: $F(2,21) = 0.942$, $p$-value $= 0.40$). The above-mentioned description attests that the three study sites had similar land use occupancy, as well as were intensively managed. We therefore consider that landscape context shaping carabid assemblages was similar between our sites and will not take it any further into account for the next analyses.

In each study site, four configurations of EIs bordering cereal crops were chosen (Fig. 1C). Two configurations displayed a single perennial EI, hedge or grassy strip, while the two others showed an association of each type of perennial EI to an annual flower strip. Annual flower strips were managed under regional AES. Installed for a minimum
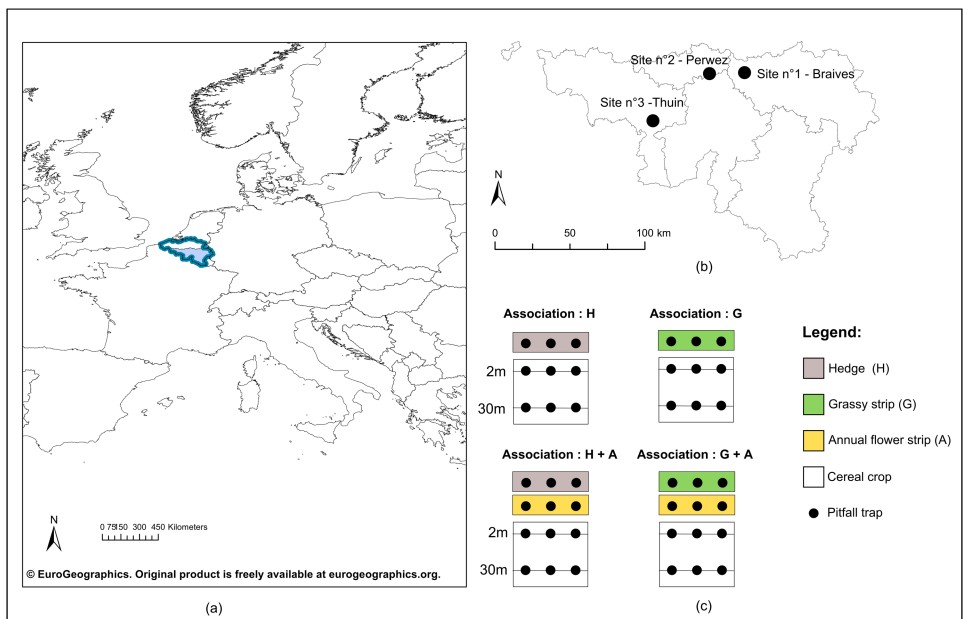

**Figure 1** **Geographical location of study sites and experimental design.** (A) Geographical emplacement in Europe of the Walloon region (in light blue) in Belgium (dark blue lines); (B) Localisation of the three study sites in Wallonia; (C) Experimental design included for each study site four configurations of ecological infrastructures (EI) at the border of crops : single hedge (H), single grassy strip (G), association of a hedge to an annual flower strip (H + A) and association of a grassy strip to an annual flower strip (G + A). Carabids were captured in pitfall traps (black dots) positionned on transect lines in all EI, as well as in crops at two and 30 m from the edge.©EuroGeographics. Original product is freely available at eurogeographics.org. Terms of the licence available at https://eurogeographics.org/products-and-services/open-data/topographic-data/.

of five years, these strips adjacent to cropped fields are ploughed every year and sown with a mix of cabbage and winter or spring cereals, to which are usually added radish, flax, buckwheat and sunflowers. The purpose of annual flower strips is to provide food resources to farmland birds during autumn and winter, as well as a refuge for small farmland fauna. Grassy strips coupled to annual flower strips were also under AES management. Sown along the annual flower strip, they were installed for a minimum of five years, normally left uncut and made of a mix of grass species (i.e., mix of reference: 45% *Dactylis glomerata*, 25% *Phleum pratense* and 30% *Festuca rubra* or 75% *Dactylis glomerata* and 25% *Festuca rubra*).

Cereal crops (predominantly wheat, but barley, oat and spelt were present as well), along with three categories of EIs (hedges, grassy strips, annual flower strips), were thus sampled. Both in crops and EIs, a transect line was drawn lengthwise the bordering EI (see Fig. 1C), on which were placed pitfall traps to collect carabids. In crops, two transects were defined: the first at two meters from the border and the second at 30 m. Traps were polypropylene cups of 8.5 cm diameter, half-filled with a mix of polyethylene glycol, diluted to 30%, and a few drops of neutral liquid soap. Their openings were flush with the soil surface, and a metal grid (heightened of 1.5 cm, with a mesh size of one cm) was used above every trap

to prevent catching rodents and other small vertebrates. Three traps were equally spaced along every transect. Traps were simultaneously open for seven days during the second week of May, June, July and at the end of September 2016. Weather conditions were similar between sites during catching sessions and in case of rain, no flooding occurred.

Carabids were identified to species level using the identification keys of *Coulon, Pupier & Queinnec (2011a)*, *Coulon, Pupier & Queinnec (2011b)* and *Roger, Jambon & Bouger (2013)*. A list of all species and their associated traits is available in Table S2.

Statistical analysis and plots were computed with R (version 3.5.1, *R Core Team, 2013*). Exploring composition of carabid communities in the different sampled configurations of EIs was performed using Principal Coordinate Analysis (PCoA). We plotted activity-density per transect. Data was standardised to take into account missing traps by averaging all individuals by traps over the whole trapping period. Then, the number of individuals from the three traps was summed to obtain an activity-density value by transect. A distance matrix of Bray-Curtis was computed from the activity-density data set and root-transformed before the PCoA calculation in order to conserve its Euclidean property.

Functional traits were used to analyse the structure of carabid assemblages. Selected traits (Table 1) were either known to be response traits of carabids to landscape change (i.e.,size and wing type, see *Pakeman & Stockan, 2014*) or inform about the potential provisioning of ecosystem service (i.e., diet) or species biology (i.e., breeding period). They were taken from the *Ribera et al. (1999)* and *Dufrêne (1992)* databases. Functional trait diversity was estimated using functional dispersion metrics (FDis index) from *Laliberté & Legendre (2010)* and the associated R package 'FD' (*Laliberté, Legendre & Shipley, 2014*). Cailliez correction for non-Euclidean distance was applied to take into account the categorical traits. Table 1 displays the trait occurrences for each EI type and crop, completed by the community weighted mean (CWM) size.

To test the influence of the type of EI and the distance to the crop edge on crop carabid assemblages, we used General Linear Models (GLM) with the 'glm' call.

This test was performed on a set of variables: carabid activity-density, species richness, functional diversity index (FDis), proportions of carabid functional traits and community weighted mean (CWM) size (see Table 1 for the traits categories). For the diet, we grouped omnivorous and granivorous species as both can exert as weed seed suppressor. We did not take into account the wing morphology since brachypterous species were rare. Distribution models for the GLMs were visually selected via the plotting of quantiles. Most variables followed a Gaussian distribution, except activity-density, granivorous carabids, large (>10 mm) and very large carabids (>15 mm) (log-transformed), and species richness (Poisson distribution). GLMs were run for each variable, testing five models: 1°EI type * Distance; 2°EI type + Distance; 3° Distance; 4°EI type and 5° the null model (~1).For each variable, we selected the best fitted model using the Akaike Information Criterion (AIC; (*Burnham & Anderson, 1998*) with a correction for small sample (AICc; (*Hurvich & Tsai, 1989*). Difference of AIC value from the best model ($\Delta$ AICc) and AICc weight (wAICc) were used to rank the models. Eventually, p-values were computed for models with a $\Delta$ AICc<2. We used a Likelihood Ratio Test (Type III Wald chisquare tests) through the command 'Anova' from the package 'car' (*Fox & Weisberg, 2011*). If significance was detected, a

Peer J

**Table 1  Proportions of selected functional traits (%) and Community Weighted Mean (CWM) size in carabid assemblages from each ecological infrastructures (EI) and from crops.** Proportions of individuals displaying the traits of interest are given by transect (averaged values for three traps).

| Type of EI | Size class (%) | | | | | Wing (%) | | | Diet (%) | | | | Breeding period (%) | |
|---|---|---|---|---|---|---|---|---|---|---|---|---|---|---|
| | 1: <5 mm | 2: 5 mm <X ≤ 10 mm | 3: 10 mm <X ≤ 15 mm | 4: >15 mm | CWM size (mm) | Brachy-pterous | Dimor-phic | Macro-pterous | Generalist predators[a] | Specialist predators[a] | Omnivorous | Granivorous | Spring | Autumn |
| H | 16.05 | 43.95 | 26.28 | 10.72 | 8.82 | 10.72 | 44.33 | 44.94 | 77.95 | 4.26 | 15.97 | 1.83 | 53.00 | 47.00 |
| G | 23.76 | 47.26 | 27.22 | 1.76 | 8.06 | 0.00 | 55.04 | 44.96 | 87.41 | 4.81 | 1.42 | 6.36 | 66.96 | 33.04 |
| A | 28.23 | 26.74 | 36.47 | 8.58 | 8.86 | 7.86 | 40.19 | 51.95 | 73.85 | 3.75 | 13.80 | 8.60 | 80.92 | 19.08 |
| C2 | 21.53 | 39.95 | 34.96 | 3.56 | 8.76 | 1.52 | 52.85 | 45.63 | 86.86 | 7.18 | 5.36 | 0.60 | 58.84 | 41.16 |
| C30 | 18.59 | 27.22 | 51.84 | 2.34 | 10.04 | 0.12 | 54.46 | 45.42 | 91.21 | 6.22 | 2.13 | 0.44 | 43.22 | 56.78 |

**Notes.**

EIs and crops are abbreviated as follow:

H, Hedges; G, Grassy strips; A, Annual flowers trips; C2, Crops (2 m from the edge); C30, Crops (30 m from the edge).

[a]'Specialist predators' refer to species feeding mainly on Collembola while Generalist predators have a wider panel of preys.

**Table 2 Diversity indices of carabid assemblages from ecological infrastructures (EI) and crops.** Mean values of carabid activity-density, species richness and FDis index per transect (sum of three traps) for each type of EI and crops. Error is given by the standard deviation.

| Type of EI | Activity-density | Species diversity | FDis |
|---|---|---|---|
| H | 73.06 ± 78.48 | 11.33 ± 4.97 | 0.25 ± 0.12 |
| G | 82.06 ± 65.21 | 14.33 ± 5.13 | 0.29 ± 0.04 |
| A | 142.11 ± 30.88 | 18.67 ± 5.13 | 0.32 ± 0.09 |
| C2 | 246.56 ± 160.81 | 16.08 ± 2.58 | 0.27 ± 0.04 |
| C30 | 269.06 ± 208.66 | 16.17 ± 2.21 | 0.27 ± 0.04 |

Notes.

EIs and crops are abbreviated as follow:

H, Hedges; G, Grassy strips; A, Annual flowers trips; C2, Crops (2 m from the edge); C30, Crops (30 m from the edge).

Tukey contrast as post-hoc test was applied to the model with the 'glht' command from 'multcomp' package (*Hothorn, Bretz & Westfall, 2008*). Residuals from significant models were checked with spline-correlograms from the package 'ncf' (*Bjornstad, 2019*) to detect patterns of spatial autocorrelation. We eventually checked for the need of a correlation structure by using GLS models, comparing the AIC of models with and without a correlation structure.

As our data did not allow to test the site effect, we provided a descriptive table of the values tested per variables (Table S3). This makes it possible to visualize whether the reported effects were consistent between the three study sites.

## RESULTS

A total of 7405 carabids, belonging to 41 species (see Annex 2) were captured during the trapping season. Of all species, 12 were represented by fewer than 10 individuals.

The first two axes of the PCoA (Fig. 2A) explained 22.70% of the variability observed in the dataset. A differentiation was observed along the first axis between crops and bordering EIs. Ecological infrastructures diverged along the second axis. Herbaceous structures (grassy and annual flower strips) and crop at 2 m) shared similarities, while communities at 30 m in the crops and in hedges were more distinct. Activity-density of carabid species characterising the crops was higher than those caught in the EIs (Fig. 2B).

Activity-density was clearly higher in ploughed structures, i.e., crops and annual flower strips, especially compared to hedges (Table 2). Species richness was fairly low in the perennial EIs compared to the annual EI and the crops. Overall, the FDis index was quite similar between the different EIs and crops, ranging from 0.25 to 0.32. The low value observed in hedges (0.25 ± 0.12) was driven by the value from the site of Thuin.

Table 1 presents the percentage of occurrence of selected traits categories and CWM size for each EI and for crops.

Small carabids (<five mm) represented between 16 and 23% of the carabid assemblages in EIs and crops, peaking to 28% in annual flower strips. Hedges and grassy strips had similar proportions of medium and large carabids (respectively around 45% and 27%), while in hedges more than 10% of caught carabids exceeded 15 mm. CWM size was identical in all EIs and at 2m from the crop border (around eight mm) but increased to 10 mm in the
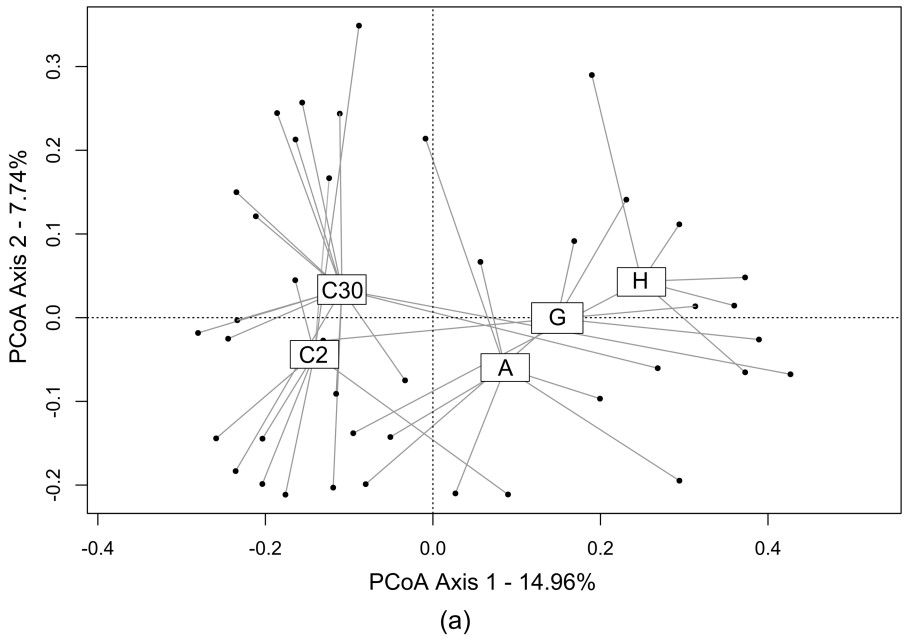

(a)

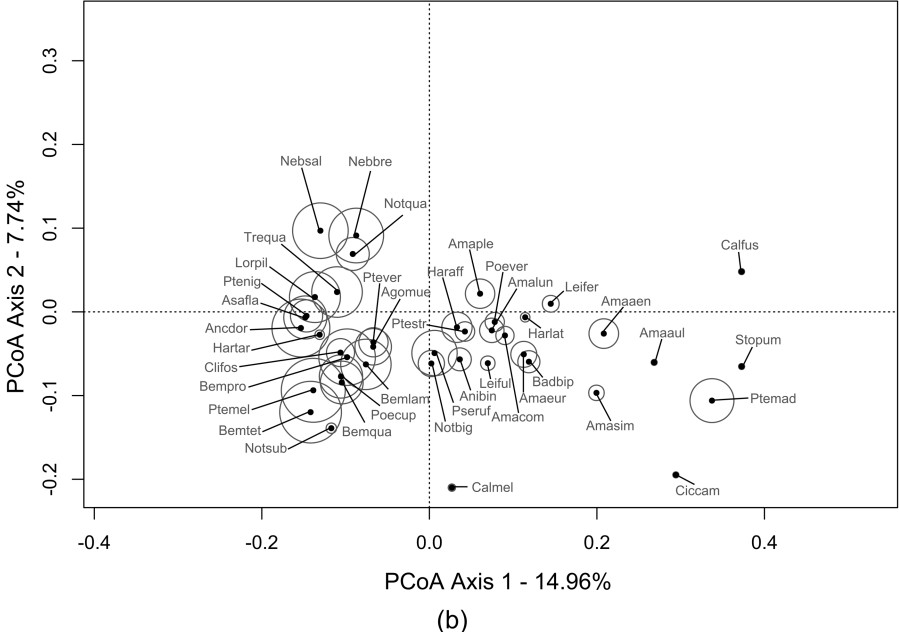

(b)

**Figure 2** **Carabid assemblages similarities between ecological infrastructures and crops by Principal Coordinate Anaysis (PCoA).** This PCoA displays the distribution of: (A) transects (i.e., sum of the three traps during all catching period) from all three study sites (grey dots). Transects are grouped under label centroids representing each type of ecological infrastructures and crops (C2 = crops (2 m from the edge); C30 = crops (30 m from the edge); A = annual flower strips; G = grassy strips; H = hedges); (B) carabid species and their activity-density, proportional to the circles size. The complete list of carabid species is available in Annex 2.

**Table 3 Results of the Generalized Linear Models applied on a selection of variables (per transect), testing both the type of ecological infrastructures and the distance from the crops' border on carabid assemblages in crops.** Models included the type of ecological infrastructure along the crop ("Type") and the distance from the crop border ("Distance"). Akaike Information Criterion corrrected for small samples (AICc), ΔAICc and AICc weight (wAICc) are given for every model. Selected models (Δ AICc < 2) are written in bold. Non-Gaussian distribution models are specified in brackets (italic) for the concerned variables.

| Variables | AICc | Δ AIC | wAIC |
|---|---|---|---|
| Activity-density *(Gaussian distribution after log transformation)* | | | |
| ~ Type * Distance | 64.2 | 20.25 | 0.000 |
| ~ Type + Distance | 50.3 | 6.35 | 0.027 |
| ~ Distance | 46.5 | 2.57 | 0.177 |
| **~ Type** | 46.8 | 2.81 | 0.157 |
| **~1** | **44.0** | **0.00** | **0.640** |
| Species richness *(Poisson distribution)* | | | |
| ~ Type * Distance | 139.8 | 18.06 | 0.000 |
| ~ Type + Distance | 129.0 | 7.30 | 0.018 |
| ~ Distance | 124.1 | 2.39 | 0.208 |
| ~ Type | 125.8 | 4.07 | 0.089 |
| **~1** | **121.7** | **0.00** | **0.685** |
| FDis | | | |
| ~ Type * Distance | −59.8 | 24.48 | 0.000 |
| ~ Type + Distance | −73.5 | 10.77 | 0.003 |
| ~ Distance | −81.8 | 2.45 | 0.222 |
| ~ Type | −76.9 | 7.36 | 0.019 |
| **~1** | **−84.3** | **0.00** | **0.755** |
| CWM size | | | |
| ~ Type * Distance | 111.1 | 12.81 | 0.001 |
| ~ Type + Distance | 101.0 | 2.72 | 0.164 |
| ~ Distance | 103.4 | 5.09 | 0.050 |
| **~ Type** | **98.3** | **0.00** | **0.641** |
| ~1 | 101.3 | 3.00 | 0.143 |
| Spring breeders | | | |
| ~ Type * Distance | 7.4 | 12.34 | 0.001 |
| ~ Type + Distance | −2.8 | 2.15 | 0.128 |
| **~ Distance** | **−3.1** | **1.82** | **0.152** |
| **~ Type** | **−5.0** | **0.00** | **0.376** |
| **~1** | **−4.8** | **0.18** | **0.343** |
| Autumn breeders | | | |
| ~ Type * Distance | 7.4 | 12.34 | 0.001 |
| ~ Type + Distance | −2.8 | 2.15 | 0.128 |
| **~ Distance** | **−3.1** | **1.82** | **0.152** |
| **~ Type** | **−5.0** | **0.00** | **0.376** |
| **~1** | **−4.8** | **0.18** | **0.343** |

**Table 3** (*continued*)

| Variables | AICc | Δ AIC | wAIC |
|---|---|---|---|
| Generalist predators | | | |
| ~ Type * Distance | −5.2 | 18.23 | 0.000 |
| ~ Type + Distance | −16.6 | 6.85 | 0.020 |
| **~ Distance** | **−22.0** | **1.44** | **0.301** |
| ~ Type | −18.8 | 4.66 | 0.060 |
| **~1** | **−23.5** | **0.00** | **0.618** |
| Specialist predator | | | |
| ~ Type * Distance | −52.8 | 24.23 | 0.000 |
| ~ Type + Distance | −65.9 | 11.13 | 0.003 |
| ~ Distance | −74.4 | 2.62 | 0.208 |
| ~ Type | −69.5 | 7.54 | 0.018 |
| **~1** | **−77.0** | **0.00** | **0.771** |
| Omnivorous *(Gaussian distribution after log transformation)* | | | |
| ~ Type * Distance | −16.4 | 18.00 | 0.000 |
| ~ Type + Distance | −28.6 | 5.77 | 0.033 |
| **~ Distance** | **−33.0** | **1.40** | **0.290** |
| ~ Type | −30.7 | 3.68 | 0.093 |
| **~1** | **−34.4** | **0.00** | **0.585** |
| Size class 1 (<5 mm) | | | |
| ~ Type * Distance | −19.1 | 10.53 | 0.004 |
| ~ Type + Distance | −26.1 | 3.60 | 0.131 |
| ~ Distance | −21.6 | 8.03 | 0.014 |
| **~ Type** | **−29.7** | **0.00** | **0.797** |
| ~1 | −24.3 | 5.41 | 0.053 |
| Size class 2 (5 mm <X ≤10 mm) | | | |
| ~ Type * Distance | −12.0 | 11.79 | 0.001 |
| ~ Type + Distance | −20.0 | 3.78 | 0.067 |
| **~ Distance** | **−23.5** | **0.20** | **0.402** |
| ~ Type | −20.5 | 3.27 | 0.086 |
| **~1** | **−23.7** | **0.00** | **0.443** |
| Size class 3 (10 mm <X ≤ 15 mm) *(Gaussian distribution after log transformation)* | | | |
| ~ Type * Distance | 8.7 | 16.10 | 0.000 |
| ~ Type + Distance | −4.5 | 2.85 | 0.098 |
| **~ Distance** | **−7.4** | **0.00** | **0.406** |
| ~ Type | −4.6 | 2.76 | 0.102 |
| **~1** | **−7.3** | **0.06** | **0.395** |
| Size class 4 (>15 mm) *(Gaussian distribution after log transformation)* | | | |
| ~ Type * Distance | −15.0 | 18.65 | 0.000 |
| ~ Type + Distance | −25.6 | 8.06 | 0.012 |
| **~ Distance** | **−31.9** | **1.72** | **0.281** |
| ~ Type | −28.2 | 5.49 | 0.043 |
| **~1** | **−33.7** | **0.00** | **0.665** |

**Table 4** **Likelihood Ratio Test calculated on the selected GLMs explaining best the variables describing carabid assemblages in crops.** Significant models ($p < 0.05$) are written in bold.

| Variables | Selected model | d.f | L.R $\chi^2$ | P value |
|---|---|---|---|---|
| Activity-density | ~1 | – | – | – |
| Species richness | ~1 | – | – | – |
| FDis | ~1 | – | – | – |
| CWM size | ~ Type | 3 | 12.64 | **0.005** |
| Spring/Autumn breeders | ~ Type | 3 | 9.03 | **0.029** |
| | ~ Distance | 1 | 0.93 | 0.335 |
| Generalist predators | ~ Distance | 1 | 1.12 | 0.290 |
| Specialist predators | ~1 | – | – | – |
| Omnivorous | ~ Distance | 1 | 1.15 | 0.283 |
| Class 1 | ~ Type | 3 | 16.09 | **0.001** |
| Class 2 | ~ Distance | 1 | 2.35 | 0.126 |
| Class 3 | ~ Distance | 1 | 2.60 | 0.107 |
| Class 4 | ~ Distance | 1 | 0.85 | 0.358 |

crop at 30 m from theborder. This is attributable to the percentage of large carabids (10–15 mm), accounting for more than 50% of all individuals caught further in the crops. Very large carabids (>15 mm) tended to be scarce in all sampled elements. Winged-phenotypes dominated in all of the EIs and in crops. Brachypterous species were rare (only two species were identified: *Pterostichus madidus* and *Stomis pumicatus*). Regarding diet, all of the EIs and crops contained more than 73% of generalist predators. If specialist species were present in similar proportions, the percentage of omnivorous was higher in hedges and annual flower strips. Granivorous carabids were more abundant in annual flower strips and grassy strips. Spring-breeding carabid species were the most abundant in EIs and crops, except at 30m from the crop edge.

Results of the GLMs are fully presented in Table 3. Tested variables (EIs and distance) did not explain the patterns of species richness, FDis index and proportion of specialist predators. For several variables, both the null model and another model had a Δ AICc <2: in that case, the model explained by the tested factor was never significant (Table 4). The type of EI modified the size and proportion of spring breeders in the crop assemblages (Table 4).The presence of a single hedge led to carabid assemblages of smaller size (Figs. 3A and 3B), with an increased proportion of spring breeders compared to an association of grassy strip/annual flower strip (Fig. 3C). Distance tended to be retained as an explanatory variable for other class sizes and diets but models were not significant. Spline-correlograms (Fig. S1) did not display spatial auto-correlation in residuals at short distances (<five km), confirming the spatial independence of transects. However, signals of potential negative auto-correlation in model residuals were present at 30 km and 70 km. Still, the addition of a correlation structure to the GLS models showed no improvement: for all three variables, GLS models with the lower AIC values are those without the correlation structure (Table 5). We thus conclude that models without correlation structure are valid.

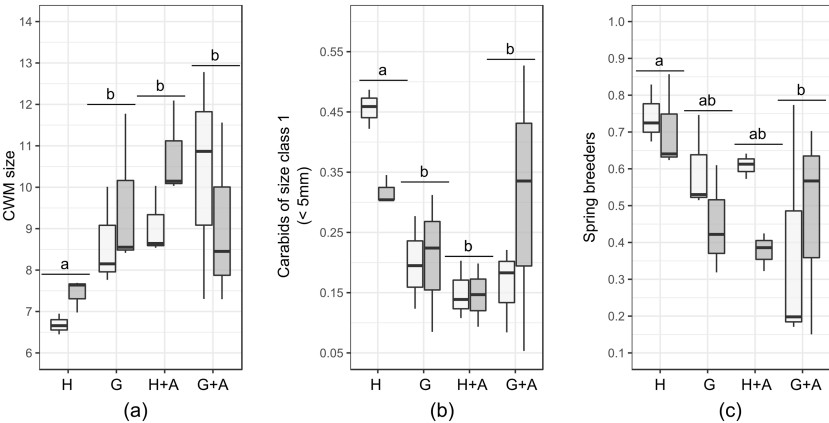

**Figure 3  Boxplots of functional traits of interest from crop carabid assemblages (by transects). Presented plots display variables that responded significantly to the type of ecological infrastructure.** Significant Linear Models (LMs) concerned: (A) Community weighted mean (CWM) size; (B) Carabids of small size (<5 mm) and (C) Proportion of spring breeders. Ecological infrastructures are abbreviated as follow : H, hedges; G, grassy strips; A, annual flower strips). Distance from the crop border is given by the shade of the boxes (light grey = 2 m from the border; dark grey = 30 m from the border). Letters above the plots indicate the results of the post-hoc Tukey test performed on the GLMs.

**Table 5  AIC values for GLS models with and without correlation structure.** All three variables (Community Weighted Mean size, Size class 1 and Spring breeders) were tested as a function of the type of ecological infrastructures.

| Correlation structure | AIC values | | |
| --- | --- | --- | --- |
| | Community Weighted Mean (CMW) Size | Size class 1 (<5 mm) | Spring breeders |
| No correlation structure | 91.62312 | −15.01437 | 5.573410 |
| corExp | 95.62312 | −11.01437 | 7.784194 |
| corLin | 95.62312 | −11.01437 | 8.829442 |
| corGaus | 95.62312 | −11.01437 | 7.225125 |
| corSpher | 95.62312 | −11.01437 | 8.829442 |
| corRatio | 95.62312 | −11.01437 | 7.695223 |

## DISCUSSION

The different sampled EIs sheltered distinct assemblages of carabids, compared to cropped fields. Regarding composition in functional traits, communities from EIs and crops were dominated by predator carabids from open land displaying a good ability to disperse, and mostly reproducing in spring. The two last traits are known to mirror the filtering initiated by landscape configuration on carabid assemblages (*Duflot et al., 2014b*), selecting highly mobile species able to cope with crop disturbances. Even in the absence of a control situation without any EI, we suspect that the landscape context largely shaped the pool of species and traits present in the agroecosystems and that local management could only slightly modulate these (*Woodcock et al., 2010*; *Trichard et al., 2013*).

Overall, sampled carabid assemblages showed a functional homogeneity, tempered by the availability of EIs. The similarity of FDis in crops bordered by different EIs suggested that even though these EIs contributed to local diversity of carabid species, their contribution to traits diversity may have been insufficient regarding the filtering exerted by the intensive agrolandscapes. When unusual traits for carabid assemblages in crops were detected, i.e., granivory, carabids were too few to raise functional diversity index (FDis), as FDis weights species by their abundance.

Still, among the tested functional traits in crops, the size of carabids was affected by the presence of single hedges. Community weighted mean (CWM) size of carabids in the crops decreased, in response to the decline of large (>10 mm) carabids and the increase of small carabids (<five mm). The distance to the crop border conversely tended to favour the proportion of large carabids. As the size of predator species plays a key role in prey-predator relationships, in terms of consumed species and predation rate (*Rusch et al., 2015*) pest control may be affected. The dominance of large carabid species is suspected to favour intra-guild predation towards other carabid species of smaller size or other potential predators may occur, possibly interfering with the desired pest control (*Prasad & Snyder, 2006*). To the contrary, according to allometric laws (*Brose, 2010*), the larger proportion of small carabid species in the presence of hedges could benefit to the consumption of small pests such as aphids.

The presence of single hedges also increased the proportion of spring breeders in the crops. This suggests that single hedges are a refuge habitat of lower quality, at least for some autumn breeding species. Poor habitat quality of hedges under anthropogenic management is common (*Magura, Lövei & Tóthmérész, 2017*) and could partly explain the questioning results of hedges, characterized by a low diversity and activity-density of carabids. This contrasts with other studies where hedges, as a part of a coherent network such as bocage landscapes, were commonly identified as distinct habitats, hosting a mix of forest and open-land species (*Fournier & Loreau, 2001*; *Duflot et al., 2014a*). This was, however, not the case on our study sites, as hedges there were planted in open-field landscapes.

Granivorous carabid species responded positively to the presence of grassy and annual flower strips adjacent to crops. As granivorous species are highly mobile in agrolandscapes to exploit food resources (*Labruyere et al., 2016*), they benefit from the establishment of weeds in EIs (*Fried et al., 2009*), from which they may feed on seeds (*Saska & Jarosik, 2001*). Still, they hardly dispersed within the crops and showed no response to the type of EI, as it has been observed by *Boetzl et al. (2018)*. Literature shows that bordering habitats have a limit of influence on crop species assemblages (*Roume et al., 2011*). Acknowledging that some species show clear preferences for either boundaries or field-edges (*Saska et al., 2007*), which may retain them from dispersing within the neighbouring field (*Boetzl et al., 2018*), the role of EIs in the enhancement of granivorous carabids in crops appeared limited in intensive agroecosystems. Moreover, if distance from the edge tends to exert a selection pressure on size, this could limit the range of granivorous carabid size diversity. As a consequence, the variety of consumed seed may be restrained as carabid size constrains seed preferences (*Honek et al., 2007*).

Though our results display clear trends, the lack of repetition of EIs per site in our study enables us to generalize them to intensive agrolandscapes. Still, our study addresses two main issues: (1) the planting of single hedges for carabids in intensive agricultural landscapes, outside any forested network and (2) the role of granivorous carabid species in the control of weed seed in crops. As single hedges turned out to be of weak support to carabids from agroecosystems, potentially impeding their movement in the landscape (*Mauremooto et al., 1995*), we encourage a deeper investigation of their role in intensive agrolandscapes. Regarding granivorous carabid species, even if the activity-density of carabids does not ensure the delivery of an ecosystem service (*Saska et al., 2008*), we suggest as an experimental perspective to investigate the presence of granivorous carabids in crops during ploughing period. As ploughing exposes weed seeds from the seed bank, it is likely an adequate time of action for their consumption by carabids if they can exert any control. Concomitantly, as distance from the border appears as a potential limiting factor to the dispersion of carabids within crops, we enjoin to investigate the spatial arrangement of EI in agroecosystems as a research perspective to promote efficient and stable pest and weed control in crops.

## CONCLUSION

Our results support that carabids can be very diverse in intensive agroecosystems (*Vanbergen et al., 2005*), though displaying a homogenisation of functional traits. Carabids fanned out in the different habitats at their disposal quite quickly, as exampled by the colonization of annual flower strips by granivorous species. Thus, adapting field edges with variate strips provide habitats and food resources to carabid species infrequent in crops or offer an adequate refuge to crop species. Differences in carabid assemblages in crops were mainly driven by the presence of hedges. Changes concerned the CWM size, which decreased in the presence of a hedge while the proportion of large carabids increased with distance from the crop border. Considering carabid assemblages and functional guilds, our results highlight that the role of ecological infrastructures in the enhancement of ecosystem services may be limited by the supported pool of species and in view of these species habitat preferences, notably for weed seed removal. To conclude, if the diversification of EIs is to be pursued in intensive agrolandscapes, their installation would benefit from a landscape-inspired planification, based on a networking of farmers' engagements to allow the development of an efficient matrix of agri-environment schemes.

## ACKNOWLEDGEMENTS

The authors thank Benjamin Daigneux who identified the sampled carabids.

### Funding

Emilie Pecheur holds a Ph.D. fellowship from the F.R.S- FNRS, whose funding covered the realization of this study. The funders had no role in study design, data collection and analysis, decision to publish, or preparation of the manuscript.

### Grant Disclosures

The following grant information was disclosed by the authors:
F.R.S- FNRS.

### Competing Interests

Julien Piqueray is employed by Natagriwal asbl.

### Author Contributions

- Emilie Pecheur conceived and designed the experiments, performed the experiments, analyzed the data, contributed reagents/materials/analysis tools, prepared figures and/or tables, authored or reviewed drafts of the paper, approved the final draft.
- Julien Piqueray and Arnaud Monty conceived and designed the experiments, contributed reagents/materials/analysis tools, authored or reviewed drafts of the paper, approved the final draft.
- Marc Dufrêne and Grégory Mahy conceived and designed the experiments, authored or reviewed drafts of the paper, approved the final draft.

### Data Availability

The raw data is available in the Supplementary Files.

### Supplemental Information

Supplemental information for this article can be found online at http://dx.doi.org/10.7717/peerj.8094#supplemental-information.

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
