# Peer review of "The influence of ecological infrastructures adjacent to crops on their carabid assemblages in intensive agroecosystems"

_PeerJ, doi:10.7717/peerj.8094_

## Round 0.1 · original submission · Major Revisions

I have now received three detailed reviews. All find your paper interesting but stress out important shortcomings. Importantly, statistical analyses need to be thoroughly revised and improved, as suggested by Reviewer 1, and model results must be included in the main text. I urge you to use a model selection approach (based on information criteria like AIC) to identify the most relevant predictor variables or test for their effect on the model’s likelihood (likelihood ratio tests). I would also like to see if model residuals show spatial autocorrelation (using acf plots or SA tests). The description of your sampling design and Figure 1 should also be substantially improved (please use colors, there are no extra charges) to better explain how you tested for differences in carabid community composition among hedges, grassy strips and flower strips (see comments by Reviewer 2). Additional important traits could be included, and you should explain which traits were assigned to which species, as requested by Reviewer 1. You could also assess the impact of different land use categories on carabid communities, as suggested by Reviewer 3. Please make sure English is thoroughly revised, all tables and figures are described in detail in the legends or titles, and the discussion of the a priori hypotheses is emphasized. I encourage you to submit a revised version.

Reviewer 1 ·

Basic reporting

The authors investigated the effects of different types of green infrastructure on carabid beetle assemblages within the infrastructure and in adjacent cereal fields. The study addresses a timely topic – many publications at the moment focus on the use of semi-natural habitats and landscape elements to facilitate ecosystem functioning. The manuscript is to the point and well written but language could profit from a thorough revision and maybe the involvement of a native speaker.

Experimental design

In general the experimental design is sound and the experiment seems to have been conducted well. However, some specific parts need more clarification (as listed below) and using only 3 real replicates per treatment could be a problem for the transferability of the results. In my comments below I try to point out how this problem could be dealt with. The statistical analysis is performed in a modern, valid way however, model result tables should be included in the main text.

Validity of the findings

The findings are interesting and appealing for the scientific peer group. However, statistics need to be revisited in order to proof that the results are sound (see major comments below). The discussion is sound but could profit from a clearer focussing on the main hypothesis and from inclusion of some newer and on the topic literature (as stated below). The conclusion is based on the results and ends in broad but precise management applications which are good but maybe at the current state not completely supported by the data.

Additional comments

Major Comments

(01) Using functional traits is timely and can bear a lot of information about the ecological importance of species distribution data (i.e. when approximating ecosystem functions). However, I am missing some very important traits which could very easily be included – specifically the community weighted mean body size (as body size is a direct proxy for food intake rates and has been shown to be a very important predictor for predation rates (Rusch et al. 2015)).
I aslo think that other traits are a bit overstressed (e.g. the four size classes) and some could be left out – especially Figure 3 would benefit greatly if not 12 separate graphs were shown but maybe only the most important ones which depict significant differences. The remaining results could be shown in a table (as more or less already done in Annex 3).
Also, the authors do not state clearly which traits were assigned to which species (size, size class, diet, wingshape …). This could be easily added in the species list (Annex 2) and is very important for replicability of the results as occasionally the same species have been assigned different traits by different authors in the past.
(02) It is unclear to me on which level the authors analysed data. The study was performed on 3 sites with 4 treatments each and 2 transects per treatment at each site. So did the authors use 3 replicates (sites) per treatment (as could be deduced from the R script)? This is a fairly low number – it could easily be increased by treating the two transects per treatment / site as replicates and accounting for pseudoreplication by using a nested random term with transect nested within treatment / site combination. The authors should clarify how they performed their analyses and calculate the alternative proposed by me as I suspect these models to have more statistical power.
(03) The authors performed GLMMs – but the current state of the manuscript does not show the results (L 163ff). Please add a table with the results of all GLMMs (fixed effect results for each of the response varibales) – similar to Annex 3 but before the Tuckey post-hoc tests were performed. Also, please include degrees of freedom (numerator, denominator).
(04) Although the authors performed a substantial literature search, some very recent and essential papers dealing with very similar or exactly the same topics are still missing (e.g. Boetzl et al. (2018), Fusser et al. (2018), Van Vooren et al. (2018)). I suggest including some of this newer publications – this could especially be interesting for the discussion.
(05) Although the results for both hypotheses are shown, the whole manuscript and especially the discussion could benefit from a stronger focus on the discussion of the a priori hypotheses.


Minor Comments
Throughout the manuscript: Please always use ‘assemblages’ instead of ‘communities’ as the term community refers to groups with different taxonomic levels (e.g. animal community in a pond) and not only a single taxonomic group (such as carabid beetles).
L 54: Replace ‘occaisioning’ with ‘causing’
L 55: ‘levels’
L 64ff: Very long and complicated sentence – please restructure and simplify (maybe through devision into two separate sentences).
L 70: ‘preserve biodiversity locally in the fields’
L 71f: Does this statement really need 4 references?
L 73f: I suggest restructuring this sentence to: ‘In the European Union, perennial elements as well as a variety of annual flower strips can be established under the framework of AES’
L 77: ‘the edge-effect’ is a bit unspecific – I guess the authors mean a positive response of carabid beetles to crop edges. However, there are edge-effects of different nature – e.g. also an edge-effect on weeds which are increased by edge proximity. I suggest to clarify this.
L 82: skip ‘spatial’
L 82ff: This sentence is unclear to me – please restructure to make the message clearer
L 86: ‘undergone radical landscape simplification or being historically intensive’
L 88: change to ‘safeguard’ – I am not sure whether the word the authors mean by ‘implementation / safeguard’ is not just ‘establishment’
L 90: In my opinion, 3 references would be enough to make a point – maybe use: (see e.g. …..)
L 93: ‘to have different carabid beetle assemblages.’
L 95: ‘traits within the’
L 100f: This sentence is lacking a main part (is rather a sub sentence of the previous one)
L 102: I do not understand what the authors mean by this sentence – please clarify.
L 114: Change to ‘In ArcMap …’
L 131f: ‘they consisted of a mixture of grass species’
L 134f: ‘were sampled.’
L 136: ‘on which pitfall traps were placed’
L 140: Metal grids: Mesh size? It might influence the data a lot as very large species get filtered out – people later using the data might want to know.
L 145: Which weeks? E.g. always the first week? Or was the time interval between trapping intervals uneven?
L 151: How was data standardized? Please explain.
L 152: ‘square root’
L 163: Which command? – It DOES make a difference sometimes …. from the R script I know that it is glmer.nb – please state it in the manuscript. (By the way: ‘Infrastructure + Parcelle+ Infrastructure*Parcelle‘ is exactly the same as ‘Infrastructure*Parcelle ‘ as the interaction also includes the fixed effects – if one wants to address only the interaction it is ‘Infrastructure:Parcelle’ – but this does not make any difference in this case).
L 175: RStudio is just a graphical interface – the authors used R which should be cited as stated when typing in ‘citation()’
L 195: This sentence is a bit clumsy – I would suggest combining the hint to the table with some information and just referring to the table in the end of the sentence.
L 202: I would suspect this is because of the use of metal grids ….
L 219: When referring to non predatory carabids, please use granivorous or spermophagous instead of herbivorous as they are not typical herbivores (not chewing on leaves for example)
L 224: replace ‘on the all’ with ‘overall’
L 228: replace ‘sheltered’ with ‘hosted’
L 230: ‘Regarding functional traits, …’
L 231: ‘predatory’
L 232: replace ‘last’ with ‘latter’
L 238: ‘to abundance’ and replace ‘engage’ with ‘cause’
L 242: replace ‘seed-eaters’ with ‘granivorous’
L 244: Please rephrase this sentence – it reads quite clumsy.
L 279: ‘benefitting omnivorous and …’ and ‘groups’
L 280: ‘species which are more sensitive to …’
L 281: ‘offered to carabids a habitat analogous to crops i.e. …’
L 286: replace ‘modified as well’ with ‘also modified’
L 290: When talking about genera, the ‘sp.’ is not needed
L 291: replace ‘What is more’ with ‘Moreover’
L 292: ‘hibernate’
L 294: ‘reproduces’
L 303: ‘in the crops’?
L 306: replace ‘seed-eaters’ with ‘granivorous’
L 307: replace ‘adapting’ with ‘enriching’ and ‘could provide’
L 311: ‘due to species specific habitat preferences’


Reference list: 70 references is a bit long for a quite short manuscript – as mentioned before there is some potential of shortening in different parts of the manuscript.
Figure 2 and 3: The placing of the letters (e.g. (a)) is a bit unusual – I would recommend the conventional upper left corner in both figures.
Annex 2: Change ‘Badister bipustulatus’ to ‘Badister bullatus’ as it is the correct name (following the databases carabidae.org or carabidfauna.org).





References

Boetzl, F.A., Krimmer, E., Krauss, J. & Steffan‐Dewenter, I. (2018) Agri‐environmental schemes promote ground‐dwelling predators in adjacent oilseed rape fields: Diversity, species traits and distance‐decay functions. Journal of Applied Ecology.
Fusser, M.S., Holland, J.M., Jeanneret, P., Pfister, S.C., Entling, M.H. & Schirmel, J. (2018) Interactive effects of local and landscape factors on farmland carabids. Agricultural and Forest Entomology, 20, 549-557.
Rusch, A., Birkhofer, K., Bommarco, R., Smith, H.G. & Ekbom, B. (2015) Predator body sizes and habitat preferences predict predation rates in an agroecosystem. Basic and Applied Ecology, 16, 250-259.
Van Vooren, L., Reubens, B., Ampoorter, E., Broekx, S., Pardon, P., Van Waes, C. & Verheyen, K. (2018) Monitoring the Impact of Hedgerows and Grass Strips on the Performance of Multiple Ecosystem Service Indicators. Environmental Management.

Reviewer 2 ·

Basic reporting

The focus of the article is interesting, especially considering the important role played by ecological infrastructures in maintaining functionally important groups of arthropods in agroecosystems. Nevertheless, there are certain issues that must be redefined and deeply improved.
First of all, I really suggest giving the manuscript to a scientific English native speaker since, as it is written now, there are many things really difficult to understand.
There are some sections that need to be restructured, like the methods, in which the statistical analysis is not clearly separated and needs to be on the whole more specific. Also the results section needs more clarity and precision.
Figure captions have to better describe the figure contents. For example, in Fig. 3 it must to be specified if boxplots are representing the estimated mean and error obtained from the GLMM.
Tables in the supplementary material do not include titles, descriptions or footnotes.

Experimental design

Sampling design is at odds with the goals of the study. It is not possible to test for differences in carabid community composition among hedges, grassy strips and flower strips since flower strips sampled are not independent infrastructures but are always associated to a hedge or a grassy strip.
The methods need to be better explained. There is a lack of information on the location of the sampled elements and consistency of their surroundings. Is the climate similar among sites? Are all the sampled elements within a site located at the same altitude? Are the elements straight? What other elements are surrounding them? Are sampled elements flat or are in slopes? What is the orientation of the elements? Is the sun exposure of the different sampled elements comparable? Are the elements straight? What is the mean length of the sampled infrastructures (mean ± SD)? All of these aspects can influence the potential results and the reader needs to know if their effects have been controlled.

Validity of the findings

Principal Coordinate Analysis is employed to test for differences in community composition between the three ecological infrastructure types, but as I mentioned above, it is not possible to evaluate if flower strips differ from the rest of typologies in their carabid composition because they are always combined with a perennial infrastructure, and do not constitute an independent type of element.
Is GLMM fitted using data collected from both the infrastructure and the crop field? I ask this because it is not possible to compare H, G, H+F and G+F combinations if you consider the whole collected data, since sampling effort differs between H/G and H+F/G+F. If you have worked only with data from crop field, you must clarify it in the text because is a crucial consideration. Additionally, there is a strong case for adding a table summarizing results obtained from applying the GLMM in the text.

Additional comments

Keywords are not provided in the manuscript.
One of the aims in this study is to compare carabid communities in different ecological infrastructure types to evaluate the potential of these elements for biodiversity conservation, so the value of carabids as a proxy of overall biodiversity should be explained in the Introduction section (L. 52: indicator of what?). Additionally, the importance of semi-natural elements or ecological infrastructures for carabid conservation should be better justified in this section.

Reviewer 3 ·

Basic reporting

Pecheur et al. studied the influence of sown annual wildflower strips on carabid beetles using pitfall traps. Thereby, AES were sown next to crop fields and adjacent hedges or grassy strips to test which combination is most beneficial for carabid communities (species richness, dispersion metrics, diet, size, wing morphology). Crops and semi-natural habitats differ in their carabid communities but both are dominated by predatory generalists. Nevertheless, hedges were poor in activity-density and changes in crop communities were mainly driven by the presence of annual flower strips. They emphasize the importance of local heterogeneity of crop edges and the low impact of hedges in poor landscapes in agroecosystems.

The manuscript is easy understandable and well written. The typical structure is given and the introduction highlights the current state of knowledge of AESs in agroecosystems. Used literature is relevant and well referenced with one exception (see general comments under line 158). Figures, raw data, and appendixes are relevant, but some details must be corrected.

Experimental design

At first, the primary research is within the scope of the journal. Methods are well described with sufficient details. Data and statistics are robust and well analyzed. However, the main limitation is the missing novelty of the study. At first, the introduction is well written and the research question is well defined. But, the second aim of the study is to discuss examples to promote ecosystem services by carabids for weed seed or pest control without testing the real potential for it (here depending on species richness and traits). Furthermore, they provide a landscape analyses based on different land use categories (crops, meadows, forests, percentage of AESs and semi-natural elements, etc.) but neither in the results nor in the discussion, they focused on that point. This however, the establishment of AESs or hedges in agroecosystems, highly depends on the surrounding landscape as they mentioned only in the conclusion at the end. Here, the authors have the data to check which impacts have different land use categories on carabid communities and in which landscapes new introduced AESs or hedges are most beneficial or not. The authors should reconsider and concrete the main aims of the study throughout the manuscript to identify the knowledge gap which is closed by their study.

Validity of the findings

see experimental design

Additional comments

Line 128: Did you measured environmental parameters of flower and grassy strips like vegetation height, coverage, soil parameter? Throughout Europe there are probably thousand kinds of AES for different taxa or aims. The audience should have a better idea how they look like here. It is an important point which you already discuss later.
Line 132: please provide also some examples for grass species of the seed mix (see comment above)
Line 145: Only one week per month can be critical. Please discuss the impact of weather (one week rain), if there was any!?
Line 158: traits were taken from Scottish carabids (Ribera 1999) and a dissertation (Dufrene 1992); better use Homburg et al. 2013: Carabids.org – a dynamic online database of ground beetle species traits (Coleoptera, Carabidae). Insect Conservation and Diversity 7. 195-205
Line 180: please provide p-values within the results and running text
Line 238: “to low”, remove one “o”
Line 251-253: Indeed, agrobiont carabids use open-ground sites during summer, but in autumn they search for overwintering sites; here hedges can fill an outstanding habitat in agroecosystems
Line 270: “Ranjha & Irmler”, switch “j & h”
Line 281-282: That is not true. The implementation and characteristics of AESs depend on so many parameters (weather, soil, land owner, seed mix, etc.). For example, in Germany there are AESs, which are completely different to crops: high and very dense vegetation etc. Please concrete that statement. See also comment under Line 128.
Line 290: “sp.” not italic
Line 399: “Pterostichus melanarius” in italic
Line 431-432: abbreviations of first names in capital letters: e.g. Holzschuh, A.
Line 528-529: species names in italic
Figure 1: change grey scale of grassy strip and annual flower strip, currently no difference visible
Figure 1: for an international audience, the map should provide an overview, where the study sites are located within Europe
Figure 3: add spaces between numbers and units in y-axis
Annex 1: change “combinaison”
Annex 1: categories (gardens, meadows, forests, etc.) have to be defined; Are forests comprise hedges or shrubs? AES? ELs?
Annex 2: change abbreviations of Ani bin and Asa fla
Annex 2: change Notiophilus biguttatus
Annex 2: add missing brackets for all species if necessary, e.g. Agonum muelleri (Herbst, 1784)
Annex 3: change abundance
Data PeerJ: change abundance
Data PeerJ: switch “,” into “.”

---

## Round 0.2 · Minor Revisions

I have now received two additional reviews for your manuscript, one that reviewed your original version and a new one. Both are positive about your manuscript and recommend acceptance provided you address some minor revisions. I agree and believe the manuscript has been greatly improved, however I still have important requests regarding the statistical analyses:

1) Table 3 is misleading. You do not run model selection across models with different response variables/distributions, so please clearly indicate which models you are comparing. Seems to me you are only presenting the best-fitting models here, in which case AIC is meaningless. Please provide a full model selection table containing all the compared models with deltaAIC <2, and provide deltaAIC and model weight. You can use the model.sel function from the MuMIn package.

2) The analysis of deviance should be replaced by likelihood ratio tests (as stated in my previous comments). You can use the drop1(model, test="Chisq") command.

I encourage you to submit a revised version.

Reviewer 1 ·

Basic reporting

No additional comments.

Experimental design

No additional comments.

Validity of the findings

With the revisions being incorporated, the analyses shown in the manuscript are conclusive and sound. No additional comments.

Additional comments

With the changes made, I beleive this manuscript improved greatly and I especially enjoyed Figure 1 (although I did not request it). With the recalculated analyses and all the essential parameters shown, I think the results are conclusive and transparent.
While this manuscrip is well written and ready for publication in the current form, I have two very quick revisions and an annotations which still need to be taken care of:


Minor comments:

L 93: "within the"
L 156: R version?
L191: A more up to date (and better) version for obtaining p-values (and also numerator and denominator Df estimated by bootstrapping methods) for mermod objects would be the Package 'lmerTest' (Kusnetzova et al. 2017)
For the problems of using 'Anova' from the 'car' package for obteining p-values from glmer, see also:
https://bbolker.github.io/mixedmodels-misc/glmmFAQ.html#what-are-the-p-values-listed-by-summaryglmerfit-etc.-are-they-reliable
('Testing hypotheses' section)
However, as p-Values usually do not differ essentially (other than degrees of freedom!) I would not insist on a re-analysis of the models. The authors stated how their models were analyzed and with the additional data provided, everybody can perform alternative, more up to date analyses if required.

Reviewer 4 ·

Basic reporting

The ms is in general well written and adds an interesting perspective regarding local scale studies on agricultural landscapes focusing on carabids, an indicator group responding both at the species and functional trait level to disturbances and supporting pest control and weed seed consumption services. Structure and literature seems sufficient as well as context. Yet, I missed support and references for a third hypothesis on your work: that competing interactions could also be occurring (lines 295-317). Author's could explore this explanations as well on discussion.

Experimental design

I was a bit confused on statistical analysis regarding the need for the use of PCoA and GLMM with the same data and purposes. Author's could elect one approach or strengthen (theoretically) the need to use the multiple statistical methodologies in the same work.

Validity of the findings

no comment

Reviewer 5 ·

Basic reporting

This paper addresses the effect of perennial ecological infrastructures on conservation of carabid species and its functional traits, often related to pest and weed control. Specifically, they compare carabidae assemblages between grassy strips, hedges and annual strips. They found a clear distinction between carabid assemblages occupying crops and ecological infrastructures (e.g. flower and grassy strips, hedges). Moreover, assemblages in ecological infrastructures were mostly dominated by generalist species and an unexpected result of low diversity and activity was found at hedges. The paper reads well and is well written, the literature used is sufficient. However, authors need to double check the whole text carefully since many mistakes are found along the paper, such as many spaces between lines, additional space between references, linked words (such as line 93 "withinthe", etc.

The parts that worries me most (described below) is regarding the sampling design, low replication and applied stats. Even though authors corrected for the problems in stats, I'm not sure it it will be enough for publication, especially due to low replication of the study.

Experimental design

There are some issues regarding sampling design and stats that concerns me. Moreover, the few replications of the study design must have implications in the discussed results and authors must acknowledge that.

Some main points that concerns me:

1- Scale of analysis:
Lines 117-118: within radii of 50 and 500 meters, distances known to influence carabid assemblages (Aviron et al., 2005). Can authors explain better? Because there is a huge difference between 50 and 500m. How does both scales affect carabid assemblages? Moreover, it is not clear what scale was used in the end (50 or 500)?

2- Functional analysis
Line 164: Functional traits were used to analyse the structure of carabid assemblages. Explain the differences and details of functional indexes. Moreover, the link between traits and service is weak.

3- Sampling independency.
*10m for independency between sampling sites* -
For me this is an important issue because not only the whole study has few replicates, but basically the samplings within sites are pseudo-replicates.

4- Adjacent crop
This is another important issue because the adjacent crop was not controlled and it will absolutely affect the carabidae assemblages. I think that the crop type must be included as predictor variable as well. Even though it was not their question, unfortunately it should have been controlled.

5 - Random effects
This is a huge problem of the paper because authors are using the Study sites (N=3) as random effects in the models, and it violates the main assumption of using random effects in mixed models, regarding the number of levels required to obtain variability (see Bolker, B. (2015). Linear and generalized mixed models. In: Ecological Statistics: contemporary theory and application. Edited by Fox, G.A., Negrete-Yankelevich, S., Sosa, V.J. Oxford University Press, United Kingdom, Pages 309-333.) Thus, study site MUST be included as fixed factor as well, otherwise it wont't be right.

6 - Temporal sampling
"Traps were simultaneously open for seven days during the second week of May, June, July and at the end of September - Not clear either. How many times each site was sampled? and when?

Validity of the findings

The low replication of the study and the lack of relationship with carabidae assemblages and ecological infrastructures might be just a sample effect due to the low replication of the study. Moreover, the analysis need to be redone, taking into account the issues I pointed before. Moreover, it is important that authors discuss the limitations of its results especially regarding the weak replication and study design.

---

## Round 0.3 · Minor Revisions

I received a late review from Reviewer 5, after you submitted your last revised manuscript. Since the review was late and the other two reviewers only requested minor revisions on your previous version, I feel reluctant to ask you to fully address all the issues raised by Reviewer 5. However, I believe this reviewer did raise some important points, and I would like to ask you to address these:

1) Please emphasize and discuss the limitations of your results considering your sample size, number of replicates and study design.
2) I requested the inclusion of ACF plots to assess spatial autocorrelation in the model residuals. The plots are not cited in the manuscript and spatial autocorrelation is never mentioned, so I agree with Reviewer 5' concerns about sampling independency.
3) Crop type should be controlled for including it as a predictor variable.
4) Study site should be included as fixed factor as well, as Reviewer 5 rightly pointed out.

---

## Round 0.4 · Minor Revisions

The manuscript has been improved but I'm not ready to accept it until you fully address the spatial autocorrelation issue raised. Spatial autocorrelation can increase Type I errors, and thus substantially modify your results. As requested previously, I would like to see ACF plots of model residuals or a test for spatial autocorrelation (like Moran's I) and gls models including a correlation structure if needed.

In addition, in line 187 you mention "Most variables followed a linear distribution". There is no "linear" distribution, there could be a linear relationshio between variables. Please rephrase this sentence and revise this paragraph to make sure you clarify which family/distribution was used for each response variable. Specify which were your response and predictor variables and simply call linear models (LM) those using a Gaussian distribution.

---

## Round 0.5 · Minor Revisions

Dear Emilie Pecheur,

corAR1 stands for "autoregressive process of order 1". This is what you use to model time series data, not spatial autocorrelation. What you need is this:

M3 <- gls(y ~ v1 + v2 + v3,
data = iph)

M4a <- update(M3, correlation = corExp(value = c(50, 0.1),
form=~ Xkm + Ykm, nugget = TRUE))
M4b <- update(M3, correlation = corLin(value = c(60, 0.1),
form=~ Xkm + Ykm, nugget = TRUE))
M4c <- update(M3, correlation = corGaus(value = c(50, 0.1),
form=~ Xkm + Ykm, nugget = TRUE))
M4d <- update(M3, correlation = corSpher(value = c(50, 0.1),
form=~ Xkm + Ykm, nugget = TRUE))
M4e <- update(M3, correlation = corRatio(value = c(50, 0.1),
form=~ Xkm + Ykm, nugget = TRUE))

AIC(M3, M4a, M4b, M4c, M4d, M4e)

Please explain this analyses in the methods, include in the result section (acf plots can go into supplementary material), and provide R scripts as supplementary material (or open-access repository).

Thanks,

Rodolfo

---

## Round 0.6 · accepted · Accept

I'm happy to accept your manuscript as you seem to have addressed all the major issues previously raised. I still found a few minor issues I kindly ask you to consider:

1) Please avoid/minimize abbreviations in Abstract, Figure and Table legends.
2) In Table 3, state "Gaussian distribution" as there is no "log distribution".
3) In Table 4, what you are presenting is Likelihood Ratio Tests, not deviance. Please correct the legend and methods (L195).